# Oxidative Stress Induced by Cortisol in Human Platelets

**DOI:** 10.3390/ijms25073776

**Published:** 2024-03-28

**Authors:** Maria Grazia Signorello, Silvia Ravera, Giuliana Leoncini

**Affiliations:** 1Biochemistry Laboratory, Department of Pharmacy, University of Genoa, 16132 Genova, Italy; mariagrazia.signorello@unige.it; 2Department of Experimental Medicine, University of Genoa, 16132 Genova, Italy; silvia.ravera@unige.it

**Keywords:** human platelets, cortisol, oxidative stress, aerobic metabolism

## Abstract

Hypercortisolism is known to affect platelet function. However, few studies have approached the effect of exogenous cortisol on human platelets, and the results obtained are conflicting and unconvincing. In this study, the effect of exogenous cortisol on several parameters indicative of oxidative status in human platelets has been analysed. We have found that cortisol stimulates ROS production, superoxide anion formation, and lipid peroxidation, with these parameters being in strict correlation. In addition, cortisol decreases GSH and membrane SH-group content, evidencing that the hormone potentiates oxidative stress, depleting platelet antioxidant defence. The involvement of src, syk, PI3K, and AKT enzymes in oxidative mechanisms induced by cortisol is shown. The main sources of ROS in cells can include uncontrolled increase of NADPH oxidase activity and uncoupled aerobic respiration during oxidative phosphorylation. Both mechanisms seem to be involved in ROS formation induced by cortisol, as the NADPH oxidase 1 inhibitor 2(trifluoromethyl)phenothiazine, and rotenone and antimycin A, complex I and III inhibitor, respectively, significantly reduce oxidative stress. On the contrary, the NADPH oxidase inhibitor gp91ds-tat, malate and NaCN, complex II and IV inhibitor, respectively, have a minor effect. It is likely that, in human platelets, oxidative stress induced by cortisol can be associated with venous and arterial thrombosis, greatly contributing to cardiovascular diseases.

## 1. Introduction

Glucocorticoids are considered part of the feedback mechanisms that reduce immune function. Since they are potent anti-inflammatory and immunosuppressive molecules, they are used to treat diseases like allergies, rheumatoid arthritis, pulmonary disorders, inflammatory diseases, organ transplantation, cancer, and other systemic diseases [1]. Moreover, glucocorticoids seem to play an important role in haemostasis [2,3,4] and can increase survival in patients affected with COVID-19 when oxygen support is required [5,6]. Cortisol, a steroid hormone, is the main human glucocorticoid produced by the adrenal cortex of the vertebrate. Cortisol is known as the stress hormone, since it mainly regulates cells’ response to various environmental and physiological stimuli such as stress [7]. Cortisol is essential for life, and it regulates or supports a variety of important cardiovascular, metabolic, immunologic, and homeostatic functions. Several studies report that cortisol increases the risk of developing thromboembolic events [8,9]. Although this hormone is known as a cardiovascular hormone, its involvement in atherosclerosis is noted, as it effects the regulation of glucose and lipid metabolism [10]. In Cushing’s syndrome, a pathological condition in which patients are chronically exposed to increased cortisol levels, a cardiovascular risk with 4-fold higher mortality than in the control population was observed [11,12]. Glucocorticoids receptors have been recently detected in platelet cytoplasm [13]. Thus, non-genomic hormones such as cortisol may alter platelet function, leading to a more rapid activation [14]. So far, some contradictory reports have been published on the effect of cortisol on platelet function. Most of them show an inhibitory effect of exogenous glucocorticoids on platelet aggregation [13,15,16,17], whereas one report indicates that hypercortisolism is associated with platelet hyperreactivity [18]. Recently, negative correlations have been observed between plasma cortisol concentration and platelet reactivity in response to arachidonic acid and ADP in older human subjects [19].

Oxidative stress is generated by unbalanced reactive oxygen species (ROS) production and antioxidant mechanisms. ROS include oxygen ions, free radicals, and peroxides, mostly derived from mitochondria and/or NADPH oxidase [20,21]. Accumulated ROS react with lipids, proteins, and DNA, causing damage leading to apoptosis and protein impairment [22,23,24,25]. Among ROS, superoxide anion and hydrogen peroxide, considered critical messengers for initiating changes in cellular signalling events [26,27], are important mediators of diseases, such as atherosclerosis [28,29], diabetes [30], and neurodegeneration [31], as well as aging [28,32]. ROS can act as second messengers [33] by regulating different signalling pathways. In addition, they are the main cause of oxidation within cells, decreasing antioxidant defences [34]. Moreover, ROS can spread away from the original site to other target sites [35], acting in cells as signalling molecules [36]. In consequence of their multiple effects they play a very important role in human pathophysiology, being involved in the promotion of cell proliferation, in the development of tumours [37], and in the process of cell death in several degenerative pathologies of the central nervous system [38].

Oxidative stress is known to be related to cardiovascular diseases (CVD) [39,40]. Recently, Karamouzis et al. [41] reported that, in Cushing’s syndrome patients, increased levels of oxidative stress markers are present. Moreover, ROS can regulate signalling pathways involved in physiological and pathological processes [22,42]. In resting platelets exposed to ROS, a significant increase in superoxide anion or hydrogen peroxide levels was observed [43,44,45,46,47]. In addition, platelets produce platelet hyperactivation and thrombosis [48]. Likely oxidative modifications of platelet proteins, such as carbonylation and oxidation of free thiol groups, can be considered important alterations of platelet macromolecules involved in hyperaggregability.

In the present paper we show that cortisol stimulates platelet aggregation and induces oxidative stress increasing ROS, superoxide anion production, and lipid peroxidation and reducing thiol groups’ content. The involvement of NADPH oxidase (Nox) and src and syk/PI3K/AKT pathways in cortisol-mediated platelet oxidative status is demonstrated. Moreover, we report that aerobic metabolism is involved in ROS formation in cortisol-treated human platelets.

## 2. Results

### 2.1. ROS Production in Platelets Treated with Cortisol

Treatment of human platelets with cortisol induces DCFH oxidation to DCF, which is considered a marker of ROS production. The dose-dependent cortisol effect is significant at 0.5 µM and peaks at 50 µM (Figure 1A,B). In Figure 1C the effect of several compounds able to inhibit oxidative stress is reported. Results obtained show that DTT, a disulphide reducing-agent, inhibits ROS production stimulated by cortisol by about 85%; 2TFP, a specific inhibitor of NADPH oxidase 1 (Nox1) [49] by about 50%; and the specific NADPH oxidase 2 (Nox2) inhibitor gp91ds-tat [50] and the cyclooxygenase inhibitor indomethacin by about 10%. To clarify which pathway could be involved in ROS formation by cortisol, the effect of various compounds able to inhibit different signal transduction pathways has been measured (Figure 1D). It was shown that cortisol-induced ROS formation is inhibited to a quite similar extent (about 50%) by the src kinase inhibitor PP2, by the syk inhibitor piceatannol, by the PI3K inhibitor LY294002, and by the AKT inhibitor MK2206. Moreover, studies on the effect of inhibitors of electron chain transport complexes involved in oxidative phosphorylation demonstrate that rotenone or antimycin A, inhibitors of respiratory chain complexes I and III, inhibit ROS production by about 70 and 90%, respectively, while malonic acid and cyanide ion, inhibitors of complex II and IV respectively, reduce ROS formation by about 15% (Figure 1E).

### 2.2. Superoxide Anion Formation in Platelets Treated with Cortisol

Cortisol induces superoxide anion increase in a dose-dependent manner peaking at 50 µM (Figure 2A,B), in strict correlation with ROS formation (y = 11.61x + 770.35, R^2^ = 0.9836).

Moreover, DTT produces an inhibition of superoxide anion formation of about 90%, 2TFP produces an inhibition of about 50%, and gp91ds-tat and indomethacin produce an inhibition of about 10% (Figure 2C). In agreement with data on ROS production (Figure 1D), PP2, piceatannol, LY294002, and MK2206 inhibit superoxide anion formation by about 50% (Figure 2D). Even the inhibitors of respiratory complexes produce an inhibition similar to that measured in ROS formation. In fact, rotenone or antimycin A inhibit superoxide anion formation by about 65% and 80% respectively, while malonic acid and cyanide ion reduce ROS formation by about 15% (Figure 2E).

### 2.3. Lipid Peroxidation Induced by Cortisol

Increased oxidative stress in cells leads to lipid peroxidation. As expected, cortisol induces lipid peroxidation in platelets. The effect is dose-dependent, peaking at 50 µM (Figure 3A,B). Lipid peroxidation is in strict correlation with superoxide anion (y = 193.04x − 65.897; R^2^ = 0.9799) and ROS formation (y = 2261x − 13.851; R^2^ = 0.9808). As shown in Figure 3C, the effect of DTT, 2TFP, gp91ds-tat, and indomethacin is superimposable on that of ROS on superoxide anion formation. The same inhibitory effect as that measured in ROS and superoxide anion formation was observed in platelets pre-treated with PP2, piceatannol, LY294002, and MK2206 (Figure 3D) or in the presence of oxidative phosphorylation complexes inhibitors (Figure 3E).

### 2.4. The Cortisol Effect on GSH and SH-Groups

As shown in Figure 1, Figure 2 and Figure 3, cortisol induces oxidative stress and, at the same time, it affects the antioxidant intrinsic properties of platelets, reducing platelet GSH content (Figure 4A) and SH-groups (Figure 4B), with the effect being dose-dependent for both parameters.

### 2.5. The Cortisol Effect on NADPH Oxidase Activity

One of the principal sources of oxidative stress in cells is the uncontrolled increase of NADPH oxidase activity. Thus, we tested the effect of cortisol on NADPH oxidase activity. We have found that cortisol dose-dependently stimulates Nox activity, peaking at 50 µM (Figure 5A,C). The obtained kinetic parameters are Km = 1.24 µM and Vmax = 17.01 µU/mg according to the related equation y = 150.15x + 121.57, R^2^ = 0.9857 (Figure 5D). Moreover, results reported in Figure 5B show that 2TFP and gp91ds-tat inhibit NADPH oxidase 1 or 2 by about 50% or 10%, respectively. DTT has an inhibiting effect of about 80%.

### 2.6. Western Blotting

Results of Figure 1D, Figure 2D, Figure 3D indicate that the src and Syk/PI3K/AKT pathways are involved in oxidative stress induced by cortisol. In fact, cortisol in a dose-dependent manner induces AKT phosphorylation (Figure 6A,B) confirming the involvement of this pathway in ROS formation.

### 2.7. Effect of Cortisol on Platelet Aerobic Metabolism

Oxidative phosphorylation can be considered one of the most important mechanisms of ROS production. Thus, some parameters indicative of platelet aerobic metabolism, such as the OCR and the ATP synthesis, have been tested. It was found that cortisol decreases OCR and ATP synthesis in a dose-dependent manner, both in the presence of pyruvate + malate (Figure 7A,B) or succinate (Figure 8A,B), respectively. The inhibiting effect appears to be higher in oxidative phosphorylation triggered by succinate, which is the more efficient substrate for platelets’ aerobic metabolism [51]. As expected, the P/O value, which is the ratio of the ATP produced to the oxygen consumed, decreases (Figure 7C and Figure 8C).

### 2.8. Effect of Cortisol on Platelet Aggregation

At last, we tested the effect of increasing concentrations of cortisol on platelet aggregation. As reported in Figure 9, cortisol induces aggregation in a dose-dependent manner. The effect is significant at 0.5 µM and peaks at 50 µM in line with results obtained in the previously tested parameters.

## 3. Discussion

The few studies that have previously approached the effect of exogenous cortisol on human platelets have produced conflicting and unconvincing results. It was reported that increased collagen induced aggregation and increased exposure of p-selectin after hydrocortisone infusion or dexamethasone treatment [52,53], but an inhibitory effect of prednisolone on platelet adhesion and aggregation [54] was shown. The interaction between cortisol and platelets probably occurs through a glucocorticoid receptor located in platelet cytosol and in platelet membrane [13].

In this paper we report that, in human platelets, cortisol stimulates aggregation and ROS and superoxide anion formation, behaving as a true agonist [43,44,45,46]. In addition, cortisol decreases GSH and SH-groups content. Thus, on the one hand, cortisol induces oxidative stress by increasing ROS formation, and on the other it depletes platelet antioxidant defences, decreasing GSH intracellular levels and oxidizing SH groups. Cortisol increases lipid peroxidation, as expected. Oxidative stress causes damage to proteins and membranes, inducing pathogenic intracellular signals with the consequence of cellular dysfunctions. The main sources of ROS in cells may be the uncontrolled increase of NADPH oxidase activity and the uncoupled aerobic respiration during oxidative phosphorylation. Human platelets express two NADPH oxidase isoforms, Nox1 and Nox2 [54], which are differently involved in platelet activation and ROS formation [55,56]. It is controversial whether Nox activity and its role in stimulating platelet activation are connected to agonists able to activate glycoprotein VI/immunoreceptor tyrosine-based activation motif (GPVI-ITAM) pathway or G-protein-coupled receptor (GPCR). Previous studies [57] have shown that Nox1 inhibition reduces collagen-induced platelet activation, ROS formation, syk phosphorylation, and thrombus formation, suggesting that Nox1 activation is strictly connected to the GPVI-ITAM pathway. Other authors [55] have reported that Nox1 has a leading function in GPCR-mediated platelet activation, whereas Nox2 is involved both in GPVI-ITAM and GPCR-dependent platelet activation exercising a dominant role in vivo in mouse platelets. Results of this study, carried out in human platelets, indicate a significant role of Nox1 in ROS formation induced by cortisol, whereas Nox2 seems to have a less important effect. ROS generated by Nox are also implicated in Syk/PLCy2 pathway activation with consequent Ca^2+^ mobilization and elevation [55,58,59]. Cortisol-induced ROS formation is partially inhibited by PP2, piceatannol, LY294002, and MK2206, indicating that the src and syk/PI3K/AKT pathways could be involved in the hormone’s effect. Moreover, cortisol stimulates phosphorylation/activation of AKT (Figure 6). Cortisol likely binds to a glucocorticoid receptor present in platelets [13] and stimulates ROS formation through Nox activation via src [60] and syk/PI3K/AKT signalling pathways. Thus, cortisol behaves as a true platelet agonist (Figure 9).

Besides NADPH oxidase activation, uncoupled aerobic respiration during oxidative phosphorylation is the other main source of ROS in cells [20]. In this study, we have shown that, in human platelets treated with cortisol, rotenone or antimycin A, inhibitors of complex I or III, respectively, greatly inhibit all tested parameters, while malonic acid and cyanide ion, inhibitors of complex II and IV, respectively, seem to be marginally implicated (Figure 1E and Figure 3E). Thus, the electron transport chain, at the level of complex I and III, seems to be involved in platelet ROS formation stimulated by cortisol. Cortisol could induce an uncoupling between the residual ORC and ATP synthesis, which, in turn, can stimulate ROS production. These effects on the aerobic metabolism are more evident when the oxidative phosphorylation occurs in the presence of succinate than in the presence of pyruvate + malate, with these results being in agreement with our previous data showing that platelet aerobic metabolism is principally driven by complexes II, III, and IV [51].

## 4. Material and Methods

### 4.1. Materials

Apyrase, bovine serum albumin, butylated hydroxytoluene, cortisol, cytochrome C, dichlorofluorescin diacetate (DCFH-DA), 5,5′-dithiobis(2-nitrobenzoic acid) (DTNB), dithiotreitol (DTT), indomethacin, leupeptin, NAD^+^, NADH, PGE_1_, piceatannol, phenylmethylsulfonyl fluoride (PMSF), PP2 analogue (PP2), protease inhibitor cocktail (Cat. N° P8340), superoxide dismutase (SOD), thiobarbituric acid (TBA), digitonin, pyruvate, malate, malonic acid, succinate, ouabain, ampicillin, di-adenosine-5′penta-phosphate, rotenone, antimycin A, ADP, and all chemicals were from Sigma-Aldrich, Saint Louis, MO, USA. Gp91ds-tat was from Selleck Chemicals, Houston, TX, USA. LY294002 was purchased from Merck Millipore, Germany. The 2(trifluoromethyl)phenothiazine (2TFP) was a gift of Prof. Bruno Tasso Dept. Pharmacy, Genoa University, Genoa. DPI, DTT, LY294002, piceatannol, PP2, and 2TFP were diluted in saline from a stock DMSO solution immediately before each experiment. ATP bioluminescence assay kit CLSII and ATP standard solution were from Roche, Switzerland.

### 4.2. Blood Collection and Preparative Procedures

Freshly drawn venous blood from healthy volunteers of the “Centro Trasfusionale, Ospedale San Martino” in Genoa was collected into 130 mM aqueous trisodium citrate anticoagulant solution (9:1). The donors claimed to have not taken drugs known to interfere with platelet function during the two weeks prior to blood collection and gave their informed consent. Washed platelets were isolated and tested immediately after blood collection. Centrifuging of fresh whole blood at 100× *g* was carried out for 25 min and 4 mU/mL apyrase and 4 µM PGE_1_ were added to the obtained platelet-rich plasma (PRP). A pellet, separated by PRP centrifugation at 1100× *g* for 15 min, was washed once with pH 5.2 ACD solution (75 mM trisodium citrate, 42 mM citric acid and 136 mM glucose), centrifuged at 1100× *g* for 15 min and then resuspended in Ca^2+^-free HEPES buffer containing 145 mM NaCl, 5 mM KCl, 1 mM MgSO_4_, 10 mM glucose, and 10 mM Hepes (pH 7.4).

### 4.3. ROS Assay

ROS assay was carried out with light modifications as reported [43,44]. Washed platelets (1.0 × 10^8^/mL), preincubated with DMSO or inhibitors for 10 min at 37 °C in the presence of 10 µM DCFH/DA, were treated with cortisol. Samples were then cooled and 2,7-dichlorofluorescein (DCF), produced by the ROS-induced DCFH oxidation, was immediately measured by flow cytometry in a Merck Millipore Bioscience Guava EasyCyte flow cytofluorimeter.

### 4.4. Superoxide Anion Assay

Superoxide anion formation was measured by the amount of reduced cytochrome C inhibited by SOD [43,61] with light modifications. Washed platelets (5.0 × 10^8^/mL), preincubated with DMSO or inhibitors for 10 min at 37 °C in the presence of 100 µM cytochrome C and 300 U SOD, if present, were stimulated with cortisol. Cooled samples were sedimented by centrifugation at 12,000× *g* for 8 min and reduced cytochrome C was measured in the supernatant using spectrophotometry at 550 nm, in a Beckman DU530 spectrophotometer, using a molar extinction coefficient of 21,100 M^−1^ cm^−1^.

### 4.5. Lipid Peroxidation Measurement

The quantification of thiobarbituric acid reactive substances (TBARS) is considered a marker of lipid peroxidation, as described [62]. Briefly, washed platelets (5.0 × 10^8^/mL), preincubated with DMSO or inhibitors for 10 min at 37 °C in the presence of butylated hydroxytoluene, were stimulated with cortisol. Then, samples were cooled in an ice bath in the presence of an equal volume of 20% trichloroacetic acid in 0.6 N HCl stopped incubation. One volume of supernatant obtained after 12,000× *g* centrifugation for 5 min was mixed with 0.2 volume of 0.12 M TBA in 0.26 M Tris (pH 7.0) and incubated for 30 min at 70 °C. The TBARS produced were assayed spectrophotometrically at 532 nm, in a Beckman DU530 spectrophotometer, with molar extinction coefficient of 156,000 M^−1^ cm^−1^.

### 4.6. GSH and Membrane SH-Group Assay

The GSH content was quantified by properly modified Tietze method [63]. Briefly, washed platelets (4.0 × 10^9^/mL) preincubated with DMSO for 10 min at 37 °C were stimulated with cortisol. Incubation was stopped by adding 0.2 M metaphosphoric acid and then the samples were centrifuged. The GSH content was determined in the supernatant and immediately mixed with 0.5 mM DTNB and 0.3 M Na_2_HPO_4_. The SH-groups were measured in the pellet resuspended in 3% SDS and then mixed with 0.5 mM DTNB and 0.3 M Na_2_HPO_4_. The GSH and the SH-groups were both quantified by spectrophotometry at 412 nm, in a Beckman DU530 spectrophotometer, with molar extinction coefficient of 13,600 M^−1^ cm^−1^.

### 4.7. NADPH Oxidase Activity Assay

The enzymatic activity of NADPH oxidase was assessed spectrophotometrically in platelet homogenates by measuring the reduction of cytochrome C at 550 nm. Briefly, washed platelets (1.0 × 10^9^/mL), added to 10 µg/mL leupeptin, 1 mM PMSF, 100 µM DTT, and 1/100 dilution protease inhibitor cocktail, were sonicated twice for 15 sec and then centrifuged at 14,000× *g* for 10 min. Aliquots of the obtained supernatant, preincubated with DMSO for 10 min at 37 °C, were treated with cortisol. Incubation was stopped by cooling samples in ice and NADPH oxidase activity was assayed as reported [43]. Protein concentration was measured with the Lowry method, using bovine serum albumin as standard protein [64].

### 4.8. Western Blotting (WB) Analysis

Washed platelets (1.0 × 10^8^/mL), preincubated with saline, were stimulated with increasing concentration of cortisol for 10 min at 37 °C. Incubation was stopped by adding 2 × Laemmli-SDS reducing sample buffer. Samples were heated for 5 min at 100 °C, then denaturing electrophoresis (SDS-PAGE) was performed on 4–20% gradient gels in which 30 μg of proteins was loaded for each sample and then transferred to nitrocellulose membranes. Running was performed in the presence of Colorburst™ Electrophoresis weight markers. Blots were blocked in 5% BSA dissolved in TBST (Tris buffer saline, pH 7.6, containing 10 mM Tris, 150 mM NaCl, and 0.1% Tween 20) at 25 °C for 1 h, and then incubated overnight at 4 °C with anti-phospho-AKT (Ser473) and anti-actin. All primary antibodies were diluted 1:2000 in TBST. Blots were extensively washed and incubated for 60 min at room temperature with horseradish peroxidase-conjugated secondary antibody (1:1000 in TBST). After further washings, blots were developed using the ECL™ system. Band density, reported as fold-change relative to control and normalized to β-actin, was directly quantified using the Bio-Rad Chemi-Doc software package (Quantity-One 4.6.6).

### 4.9. Oximetric Analysis

The oxygen consumption was measured at 37 °C with an amperometric O_2_ electrode in a closed chamber (Unisense, DK). Washed platelets (1.0 × 10^9^/mL) were preincubated with DMSO and then treated at 37 °C with cortisol directly in the closed chamber. From each sample, 10 µg of total proteins, determined by Bradford assay [65], was resuspended in a medium containing 137 mM NaCl, 5 mM KH_2_PO_4_, 5 mM KCl, 0.5 mM EDTA, 3 mM MgCl_2_, and 25 mM Tris, pH 7.4 and permeabilized with 0.03% digitonin for 10 min. To stimulate the Complexes I, III, and IV, 10 mM pyruvate and 5 mM malate in the presence of 0.2 mM ADP were added, while 20 mM succinate in the presence of 0.2 mM ADP was used to activate the Complexes II, III, and IV [51].

### 4.10. Fo-F1 ATP Synthase Activity Assay

Washed platelets (1.0 × 10^9^/mL) were preincubated with DMSO or inhibitors for 10 min at 37 °C and stimulated with cortisol. Incubation was blocked by cooling samples in ice, then 10 µg of total proteins, determined by Bradford assay [65], of each sample was added to the incubation medium (0.1 mL final volume), containing 10 mM Tris (pH 7.4), 50 mM KCl, 1 mM EGTA, 2 mM EDTA, 5 mM KH_2_PO_4_, 2 mM MgCl_2_, 0.6 mM ouabain, 0.040 mg/mL ampicillin, 0.2 mM di-adenosine-5′penta-phosphate, and the metabolic substrates 10 mM pyruvate + 5 mM malate or 20 mM succinate. Washed platelets were warmed for 10 min at 37 °C, then ATP synthesis was induced by the addition of 0.1 mM ADP. The ATP synthesis was measured using the luciferin/luciferase ATP bioluminescence assay kit CLSII, on a Promega GloMax^®^ 20/20 Luminometer. ATP standard solutions were used in the range 10^−10^–10^−7^ M for calibration [51].

### 4.11. P/O Ratio Calculation

The P/O value is the ratio of the nmol of ATP produced to the nmol of oxygen consumed. In coupled conditions, when the oxygen consumption is completely associated with the ATP synthesis, the P/O ratio is about 2.5 or 1.5 in the presence of pyruvate + malate or succinate, respectively [66]. In the uncoupled status, this ratio decreases, being in correlation with the degree of the oxidative phosphorylation inefficiency.

### 4.12. Aggregation

Platelet aggregation was performed in a Bio-Data Aggregometer according to Born’s method [67]. Briefly, washed platelets (3.0 × 10^8^/mL) were preincubated with saline for 3 min before platelet stimulation with cortisol. Platelet aggregation was quantified by the light transmission reached within 6 min at 37 °C.

### 4.13. Statistical Analysis

Data are mean ± SD of at least four independent experiments, each performed in duplicate. Statistical comparisons between two groups were performed through the unpaired *t*-test. To compare multiple groups, one-way ANOVA and Tukey’s post hoc test were used. Statistical significance was defined as *p* < 0.05.

## 5. Conclusions

In conclusion, in human platelets, cortisol stimulates aggregation and induces oxidative stress, increasing ROS, superoxide anion levels, and lipid peroxidation. At the same time, cortisol depletes platelets of their physiological antioxidant defences by decreasing GSH content. ROS and superoxide anion production is mainly due to the NADPH oxidase activation and to the uncoupled mechanisms between OCR and ATP synthesis. These effects seem to be mediated by src and syk/PI3K/AKT signalling pathways. Hence, the dysfunctional aerobic metabolism produced by platelet treatment with cortisol leading to oxidative stress could be associated with venous and arterial thrombosis, greatly contributing to cardiovascular diseases.

## Figures and Tables

**Figure 1 ijms-25-03776-f001:**
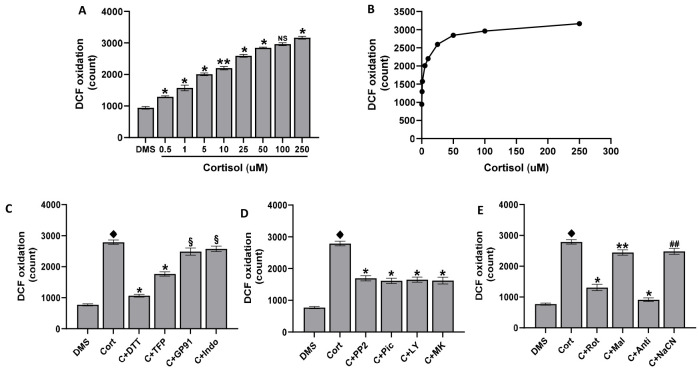
ROS formation. Washed platelets (1.0 × 10^8^/mL), loaded with 10 µM DCFH-DA and preincubated with DMSO (DMS), were treated for 15 min with cortisol as indicated (panels (**A**,**B**)). In panels (**C**–**E**), washed platelets (1.0 × 10^8^/mL), loaded with 10 µM DCFH-DA and preincubated for 10 min with DMSO (DMS), 100 µM DTT, 10 µM 2-TFP (2TFP), 50 µM gp91ds-tat (GP91), 10 µM indomethacin (indo), 10 µM PP2, 30 µM piceatannol (Pic), 20 µM LY294002 (LY), 20 µM MK2206 (MK), 1 µM rotenone (Rot), 5 mM malonic acid (Mal), 10 µM antimycin A (Anti), and 0.5 M NaCN (NaCN) were stimulated for 15 min with 50 µM cortisol. At the end of incubation, samples were immediately analysed by flow cytometry. Data are the mean ± SD of at least four experiments carried out in duplicate. One-way ANOVA and Tukey’s post hoc test are shown in panels (**A**,**B**): * *p* < 0.0001, ** *p* < 0.0005, NS, not significant. Unpaired *t*-test is shown in panels (**C**–**E**): ♦ *p* < 0.0001 vs. DMS; * *p* < 0.0001, ** *p* < 0.0005, ## *p* < 0.005, § *p* < 0.01 vs. cort.

**Figure 2 ijms-25-03776-f002:**
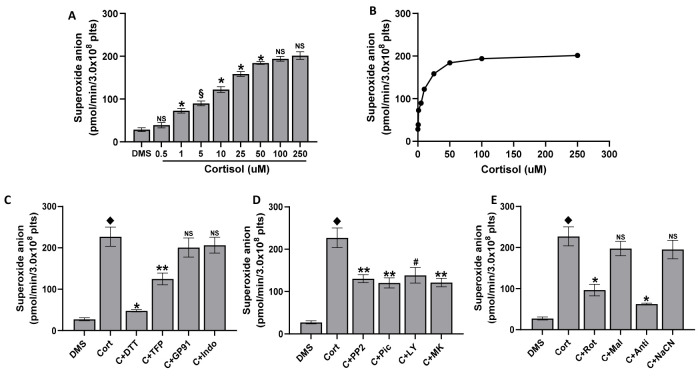
Superoxide anion formation. Washed platelets (5.0 × 10^8^/mL), preincubated with DMSO (DMS) in the presence of 100 µM cytochrome C and 300 U SOD, if present, were stimulated for 15 min with cortisol as indicated (panels (**A**,**B**)). In panels (**C**–**E**), washed platelets (5.0 × 10^8^/mL), preincubated for 10 min with DMSO (DMS), 100 µM DTT, 10 µM 2-TFP (2TFP), 50 µM gp91ds-tat (GP91), 10 µM indomethacin (indo), 10 µM PP2, 30 µM piceatannol (Pic), 20 µM LY294002 (LY), 20 µM MK2206 (MK), 1 µM rotenone (Rot), 5 mM malonic acid (Mal), 10 µM antimycin A (Anti), and 0.5 M NaCN (NaCN) in the presence of 100 µM cytochrome C and 300 U SOD, if present, were then treated for 15 min with 50 µM cortisol. The cytochrome C reduction was measured by spectrophotometry, as detailed in Methods. Data are the mean ± SD of at least five experiments carried out in duplicate. One-way ANOVA and Tukey’s post hoc test relate to panels (**A**,**B**): * *p* < 0.0001, § *p* < 0.01, NS, Not Significant; Unpaired *t*-test relates to panels (**C**–**E**): ♦ *p* < 0.0001 vs. DMS; * *p* < 0.0001, ** *p* < 0.0005, # *p* < 0.001, NS, Not Significant vs. cort.

**Figure 3 ijms-25-03776-f003:**
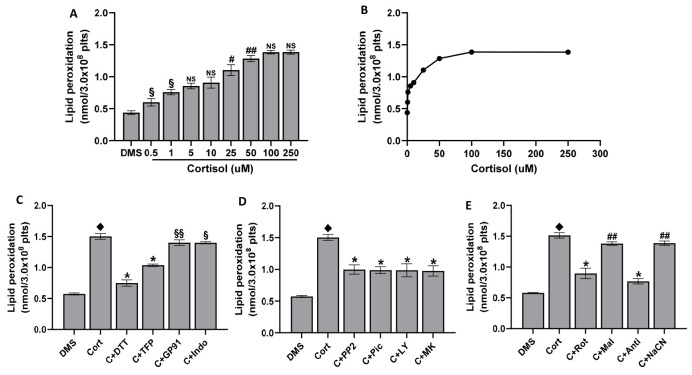
Lipid peroxidation. Washed platelets (5.0 × 10^8^/mL) were stimulated for 15 min with cortisol, as indicated (panels (**A**,**B**)). In panels (**C**–**E**), washed platelets (5.0 × 10^8^/mL), preincubated for 10 min with DMSO (DMS), 100 µM DTT, 10 µM 2-TFP (2TFP), 50 µM gp91ds-tat (GP91), 10 µM indomethacin (indo), 10 µM PP2, 30 µM piceatannol (Pic), 20 µM LY294002 (LY), 20 µM MK2206 (MK), 1 µM rotenone (Rot), 5 mM malonic acid (Mal), 10 µM antimycin A (Anti), and 0.5 M NaCN (NaCN), were treated for 15 min with 50 µM cortisol. Lipid peroxidation was quantified by measuring thiobarbituric acid reactive substances (TBARS), as detailed in Methods. Data are the mean ± SD of at least five experiments carried out in duplicate. One-way ANOVA and Tukey’s post hoc test relate to panels (**A**,**B**): # *p* < 0.001, ## *p* < 0.005, § *p* < 0.01, NS, Not Significant; Unpaired *t*-test relates to panels (**C**–**E**): ♦ *p* < 0.0001 vs. DMS; * *p* < 0.0001, ## *p* < 0.005, § *p* > 0.01, §§ *p* < 0.05, NS, Not Significant vs. cort.

**Figure 4 ijms-25-03776-f004:**
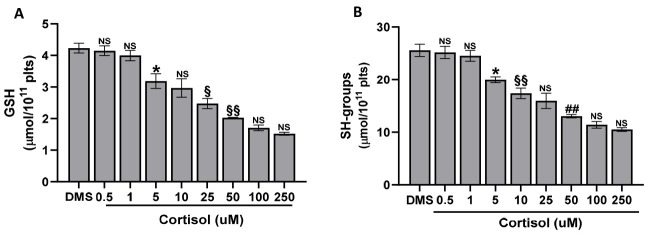
GSH and SH-groups levels in human platelets. Washed platelets (4.0 × 10^9^/mL), preincubated with DMSO (DMS), were stimulated for 15 min with cortisol, as indicated. At the end of incubation, samples were precipitated by metaphosphoric acid and GSH or SH-groups quantified as detailed in Methods. Data are the mean ± SD of at least four experiments carried out in duplicate. One-way ANOVA and Tukey’s post hoc test relate to panels (**A**,**B**): * *p* < 0.0001, ## *p* < 0.005, § *p* < 0.01, §§ *p* < 0.05, NS, Not Significant.

**Figure 5 ijms-25-03776-f005:**
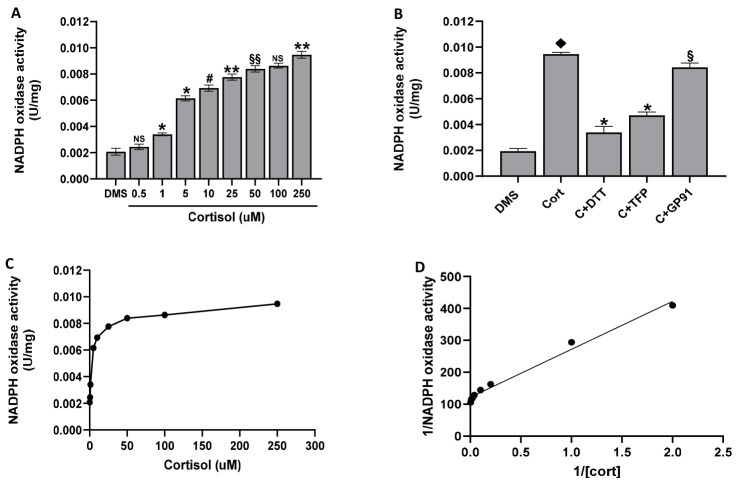
NADPH oxidase activity. Washed platelets (2.0 × 10^9^/mL), preincubated with DMSO (DMS), were treated with cortisol as indicated (panel (**A**,**B**)) or preincubated for 10 min with DMSO (DMS), 100 µM DTT, 50 µM gp91ds-tat (GP91), and 10 µM 2-TFP (2TFP) and then stimulated for 15 min with 50 µM cortisol. At the end of incubation, samples were sonicated and centrifuged. NADPH oxidase activity was assayed in the supernatant, as detailed in Methods. Data are the mean ± SD of at least six experiments carried out in duplicate. One-way ANOVA and Tukey’s post hoc test relate to panels (**A**,**B**): * *p* < 0.0001, ** *p* < 0.0005, # *p* < 0.001, §§ *p* < 0.005, NS, Not Significant; Unpaired *t*-test relates to panels (**C**,**D**): ♦ *p* < 0.0001 vs. DMS; * *p* < 0.0001, § *p* > 0.01 vs. cort.

**Figure 6 ijms-25-03776-f006:**
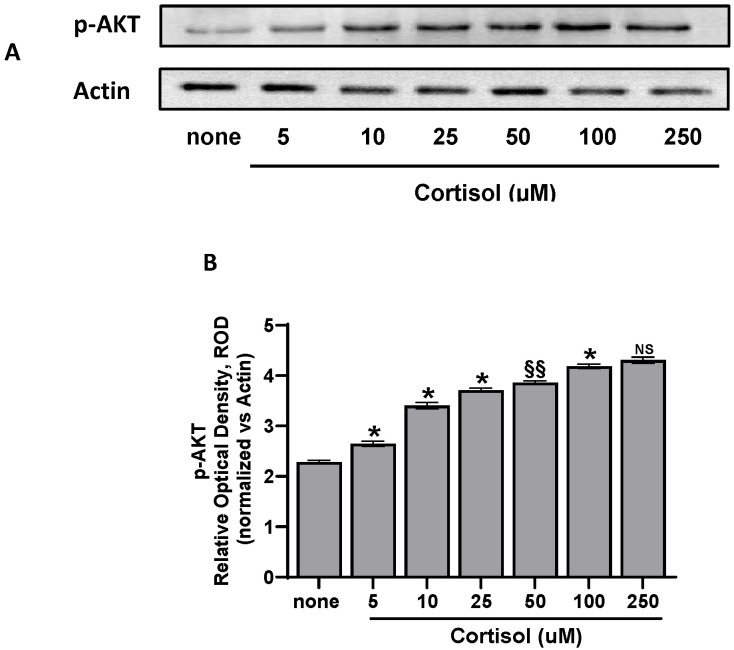
The cortisol effect on AKT phosphorylation. Washed platelets (1.0 × 10^8^/mL), preincubated with saline, were stimulated for 10 min at 37 °C with cortisol, then treated for Western blot analysis. Panel (**A**) shows the WB signals; panel B shows the densitometric analysis of signals expressed as relative optical density (ROD), normalized against the actin signal, used as housekeeping protein. Data in panel (**B**) are the mean ± SD of at least three experiments. One-way ANOVA and Tukey’s post hoc test: * *p* < 0.0001, §§ *p* > 0.05, NS, not significant.

**Figure 7 ijms-25-03776-f007:**
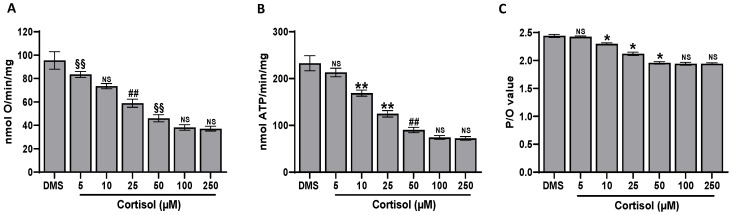
Aerobic metabolism measured in the presence of pyruvate + malate. Washed platelets (1.0 × 10^8^/mL) preincubated with DMSO (DMS), were stimulated for 1 min at 37 °C with cortisol. Panels (**A**–**C**) show the oxygen consumption, the aerobic ATP synthesis, and the P/O ratio, respectively. Data are the mean ± SD of at least three experiments. One-way ANOVA and Tukey’s post hoc test: * *p* < 0.0001, ** *p* > 0.0005, ## *p* < 0.005, §§ *p* < 0.05. NS: not significant.

**Figure 8 ijms-25-03776-f008:**
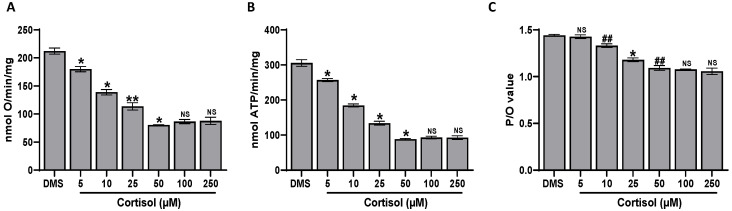
Aerobic metabolism measured in the presence of succinate. Washed platelets (1.0 × 10^8^/mL), preincubated with DMSO (DMS), were stimulated for 1 min at 37 °C with cortisol. Panels (**A**–**C**) show the oxygen consumption, the aerobic ATP synthesis, and the P/O ratio, respectively. Data are the mean ± SD of at least three experiments. One-way ANOVA and Tukey’s post hoc test: * *p* < 0.0001, ** *p* < 0.0005, ## *p* < 0.005. NS: not significant.

**Figure 9 ijms-25-03776-f009:**
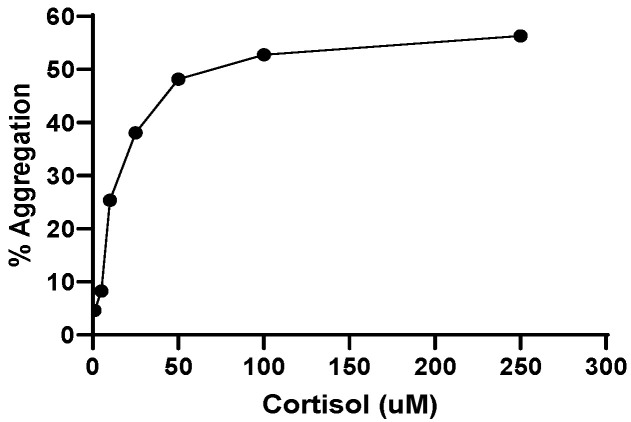
Cortisol-induced platelet aggregation. Washed platelets (3.0 × 10^8^/mL) preincubated at 37 °C in the presence of DMSO were challenged with cortisol, as indicated. Platelet aggregation was monitored as described in Methods and quantified by the light transmission reached within 6 min. Tracing is representative of four independent experiments.

## Data Availability

Data contained within the article.

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
