# Peer review of "Oxidative Stress Induced by Cortisol in Human Platelets"

_ijms, 2024, doi:10.3390/ijms25073776_

Round 1

Reviewer 1 Report (Previous Reviewer 2)

Comments and Suggestions for Authors

The revision has improved the manuscript. The authors have adequately addressed my concerns. I have no further comments.

Reviewer 2 Report (Previous Reviewer 3)

Comments and Suggestions for Authors

The authors have satisfactorily addressed my concerns.

This manuscript is a resubmission of an earlier submission. The following is a list of the peer review reports and author responses from that submission.

Round 1

Reviewer 1 Report

Comments and Suggestions for Authors

1. Lack of medical characteristics of blood donors (lack of basic biochemical analyzes of serum and morphology proving that they are actually healthy people, gender and age of volunteers were not provided). Some potentially important information was not provided, such as whether the subjects were smokers or non-smokers. It is unclear whether platelets were isolated and tested immediately after blood collection, or whether they were stored as platelet concentrates in a blood bank and made available to researchers, perhaps after their expiration date for transfusion. Because platelets are very sensitive to shear stress, needles of appropriate diameter should be used to collect blood, so that the platelets do not pass through a too narrow channel during blood sample. The lack of such technical data significantly reduced the scientific value of the article.

2. TBARS is an archaic and very non-specific method. Lipid peroxides are known to be generated by thiobarbituric acid itself during the reaction. Modern biochemistry has better and more specific methods for testing the concentration of lipid peroxides, such as HPLC, also applicable to blood platelets.

3. Judging by the description, the GSH determination method is also not specific and does not allow for the assessment of the concentration of GSH only, but of thiols in general.

Since the Authors did not describe the procedure of separattion of platelet membranes, it does not seem that the -SH concentration in platelet membranes was actually determined. Since the whole platelets were resuspended in SDS, the SH concentration was assessed in the entire platelet lysate, and not only in the membranes, after all membranes were not isolated.

4. A non-specific method was used to assess NADPH oxidase activity.

5. The statistical analysis used raises the greatest concerns. It was not done correctly. Therefore, the conclusions drawn cannot be considered reliable. It was not checked whether the obtained data were normally distributed and whether the variances were homogeneous. Therefore, it should be assumed that the described selection of statistical tests (two-way ANOVA, u M-W, t-Student) tests was carried out arbitrarily, without checking the assumptions. The use of the ANOVA test and the U M-W test following ANOVA as post-hoc test is a complete confusion of concepts because the Authors performed a parametric test (ANOVA) and then performed a non-parametric test (U M-W). Furthermore, the U M-W test cannot be used for multiple comparisons. Judging by the description, a paired Student's t-test should be used in panels C-E.

6. If the U M-W test was actually used (if there were statistical reasons for it), then the means with SD should not be presented in the figures.

7. The control should not be saline, since cortisol does not dissolve in physiological saline, but in organic solvents (DMSO, ethanol, etc.), similarly to most (if not all) other pharmacological modulators used in the study. Only cortisol salts intended for clinical use dissolve well in water. Therefore, the solvent control used is inapropriate, making all experiments unreliable.

Due to basic errors such as incorrect statistical analysis, non-specific and archaic methods, and poor description of experiments, I recommend rejection of the submitted manuscript without the possibility of resubmission.

Reviewer 2 Report

Comments and Suggestions for Authors

In this study, the authors evaluated the effect of exogenous cortisol on several oxidative stress parameters in human platelets. Cortisol increases ROS production and reduces the antioxidant defense of platelets. The authors proposed the involvement of an uncontrolled increase in NADPH oxidase 1 activity in the mechanism of cortisol-induced ROS formation.

Finally, the authors hypothesize that cortisol-induced oxidative stress in platelets may be associated with the occurrence of cardiovascular events.

The study is interesting, clearly written and the authors performed several analyses to support the role of cortisol-induced oxidative stress in platelets.

I have some considerations:

1. In the discussion, the authors state that there is no involvement of NOX2 in the production of platelet oxidative stress. However, many studies indicate the important role of NOX2 in platelet activation and ROS production (https://doi.org/10.1111/jth.14240, doi: 10.1161/ATVBAHA.116.307308, 10.1161/CIRCULATIONAHA.112.000966; doi: 10.1161/ATVBAHA.110.217885, doi: 10.1152/ajpheart.00799.2009). Moreover, the study by Delaney et al (ref 60) showed that platelet ROS generation was defective in both NOX1(−/y) and NOX2(−/−) knockouts. Therefore, this point should be better discussed and investigated.

2. In accordance with the previous point, the authors should also test the role of NOX2 by using a specific inhibitor (gp91 ds-tat or GSK2795039) to show an exclusive role of NOX1 in this pathway.

3. In all presented results, the lower concentration of 5 mM of cortisol had a significant effect. Did the authors also try lower concentrations? In patients after a cardiac event, for example, the cortisol level was 16.8 µg/dl. Therefore, the concentrations effectively tested are very high.

4. Please provide a better actin image in western blot analysis in Figure 6.

Reviewer 3 Report

Comments and Suggestions for Authors

In this manuscript the authors evaluate the effect of cortisol on platelet function as measured by ROS production.  While some of the data is interesting, the authors need to actually show that cortisol treatment influences platelet activation.  

Does cortisol treatment alone cause platelet aggregation?  The signaling pathway the authors evaluated contains a lot of commonality with signaling from receptors for physiological agonists. 

Why are platelet concentrations so different from assay to assay?

What receptor is cortisol binding on the platelet surface?  Does that receptor signaling through Src/Syk/Akt?  This was not clear in the manuscript.

Comments on the Quality of English Language

The manuscript is full of grammatical errors.  In some instances the grammatical errors make it very difficult to discern the authors' message.